# The Issue of Rights of Religious Freedom in Some Domestic Violence Cases in Indonesia

**Lidwina Inge Nurtjahyo**

Law, Society, and Development, Faculty of Law, Universitas Indonesia, Depok 16424, Indonesia; lidwina.inge@ui.ac.id

**Abstract:** Based on the National Commission for the Protection of the Rights of Women and Children of Indonesia's annual report, in 2020 there were 11,105 cases of domestic violence reported. Those domestic violence cases were caused by complex factors. One of the causes is the limitation of religious freedom in the family. In Indonesia, between 2010 and 2019, there were several cases of domestic violence caused by women choosing different religions from their parents or husband. Domestic violence involving limitation of the rights of religious freedom is sometimes resolved by divorcing or by completing it with coercive efforts. The rights of religious freedom in Indonesia, although protected by the Constitution and by the Act of Protection of Human Rights No. 39 of 1999, still face various challenges in implementation. The choice of religion in some families is highly influenced and determined by the authority in the family. This article analyzes the secondary data from online news, verdicts, and statistics from the Supreme Court Directory between 2010 and 2019. Findings are analyzed using the perspective of gender studies and anthropology of law.

**Keywords:** the rights of religious freedom; domestic violence against women; gender and law

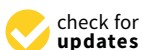



## 1. Introduction

The freedom to have religion is part of a person's rights as a human being. This right is inherent from the time the human is born. The Universal Declaration of Human Rights (UDHR) contains the recognition statement in Article 18 as follows:

"Everyone has the right to freedom of thought, conscience, and religion; this right includes freedom to change his religion or belief, and freedom, either alone or in community with others and in public or private, to manifest his religion or belief in teaching, practice, worship, and observance".

The Universal Declaration of Human Rights article states that a person's rights not only include freedom of thought, conscience, and to have a religion or practice it, but also to change his or her religion or belief. As such, the UDHR recognizes a person's right to change one's faith in his or her spiritual journey.

The spirit of the UDHR in the Indonesian context is supported by the Constitution of the Republic of Indonesia; in particular, Article 28 E and Article 29. Article 28 E of the Amended Constitution of the Republic of Indonesia stipulates that Indonesian citizens have the right to embrace a religion and worship according to that religion. They also have the right to obtain education, citizenship, and a place to live in all regions of Indonesia, including upon changing their residence.

The recognition of rights from the perspective of citizens is guaranteed by the State in Article 29 Paragraph (2). In this article, it is stipulated that the State guarantees the freedom of every citizen to embrace a religion. The State also guarantees citizen's rights to worship according to that religion and belief.

The principle of freedom to embrace a religion, worship, and even to change one's belief, as recognized by the state, in practice cannot always be implemented. In some cases, someone's freedom to choose their religion, practice, or convert his/her religion will be met with rejection from the family and society. The issue of rejection from the family

could be in various forms, ranging from verbal violations to psychological abuse, and even physical violence. In situations where the individual who converts to another religion that is different from the family's religion is a woman or a child, violence often occurs. The National Commission on Violence Against Women notes that domestic violence could happen when women or children choose to convert their beliefs. In its annual notes, the National Commission on Violence Against Women even states that in 2020 there were 11,105 cases of domestic violence reported. Those domestic violence cases were caused by various factors. One of the causes is the issue of religious freedom in the family.

Changes in belief by one family member, especially women, often lead to domestic violence. To find the relation between a person's changing beliefs and the occurrence of domestic violence, in the Indonesian context, the relation between the convert and domestic violence can be found in some divorce cases at the courts. In divorce cases, both processed in the district and religious courts, we can find from the judge's decisions several clues or indicators of domestic violence that was caused by one party's conversion to another religion or belief. What indicators or keywords can be used to find the indications of domestic violence are described in the Methods section of this paper.

## 2. Research Problems

This paper tries to explain that restrictions on the right to freedom of religion with various ideological and cultural arguments have occurred in the family in the Indonesian context. Restrictions on religious freedom in the family are not easy to disclose to the public. However, in some judge's decisions related to divorce cases, this limitation can be disclosed, with the forms of domestic violence that accompany it.

Why is divorce carried out in a civil court in the Indonesian context more of an option to stop domestic violence experienced by one of the parties to a marriage (including restrictions or coercion to embrace a certain religion)? Because the Indonesian people, especially women, are reluctant to deal with processes in criminal law that are considered tiring and longer because they deal with multilevel examinations (Wulandari 2020). Bartky (2005) even stated that women in particular often feel afraid to face the law because the legal language is considered complicated and intimidating.

This paper describes how the practice of freedom of religion and worship in Indonesia, especially in the family, is influenced by the construction of power relations within the family. In some cases, found from online news and court decisions which are discussed in this article, the issues of power relations, gender construction, and religious rights issues are intertwined.

## 3. Methods

This article was written using secondary materials. Collection of these secondary materials was carried out in several ways and included several types. First, the author collected and analyzed laws and regulations. Second, the author analyzed online media news. The news that was selected was that containing the issue of domestic violence and differences and/or changes in belief. Third, the author also used data obtained from government and non-government institutions related to cases of domestic violence, divorce, and causes of divorce. Fourth, the author also analyzed the nine court verdicts on divorce cases.

Both online news and judges' decisions, as well as collected data from government agencies and NGOs, originated from the period 2010–2020. The time limitation for documents or news between 2010 and 2020 is intended not only to help focus the search but also to see developments related to the practice of freedom of religion and worship in the past 10 years in Indonesia, especially in the family sphere.

All these materials were selected by tracing them using three keywords. Those keywords were divorce, domestic violence, and apostasy. Selection of cases in the news media focused on cases of domestic violence that befell women and/or children due to

changes in beliefs held by women and/or children. The regulations, cases from online news, and court verdicts are from Indonesia.

Court decisions were selected from the Directory of Verdicts of the Republic of Indonesia's Supreme Court. These verdicts are court decisions in civil cases, particularly divorce cases. Six cases were selected. Three cases were decided by the religious court and the other three were decided by the district court. Cases that proceed by the religious court are divorce cases filed by the couple who register their marriage at the religious affairs office, especially for those who are married according to Islam. A divorce case that proceeds by the district court is a divorce case filed by a married couple who registered their marriage at the Civil and Population Registration Office. Marriages that are registered at the Civil and Population Registration Office are usually marriages that are not based on Islamic law.

The choice of three cases from religious court decisions and three other cases from district court decisions was made to map the variation in cases. These variations in the cases are:

- One of the parties filed a divorce suit because she/he experienced violence after changing her/his belief to a new religion.
- One of the parties filed a divorce suit because he/she was forced to follow the other's belief and experienced violence because he/she refused to convert.
- One of the parties returned to their original belief before marriage and became the perpetrator or victim of domestic violence because of this decision.

The material for this paper was obtained through several stages of research. First, online news searches about cases of domestic violence. To collect news about domestic violence caused by one party changing religion, several keywords were used as filters. Those keywords were violence against wife/husband/children, and change of religion/belief. Then, to prevent the use of one-sided news, for each case found to be related to domestic violence because one of the parties changed religion, two or more news stories from different media were selected.

The second step was analyzing the Supreme Court decision. The material for analysis in this paper is some of the judges' decisions. The judge's decision used was the judge's decision produced at the Supreme Court level. The decisions were selected using a series of keywords as filters. The keywords used in the searching process were civil law cases, divorce, domestic violence, change of religion/belief, and apostasy

The process of browsing the Directory of Supreme Court Decisions is not just a googling activity that is easily carried out. Knowledge is needed to understand what kind of decisions are needed to be analyzed. In filtering the desired decision, it took several keyword changes until finally a combination of civil, divorce, apostasy, conversion, and domestic violence was found. Approximately 86,039 decisions that were found to contain a combination of civil, divorce, and apostasy, then filtered again using the classification of occurrence of domestic violence. It turns out that not all decisions also had complete case files, and of course this made the analysis process difficult.

Of those 86,093 cases, after being screened, 107 cases fulfilled the criteria, containing elements of apostasy and domestic violence. However, out of 107 cases, only 80 had complete file attachments. From those 80 cases, after the researcher reading the case files one by one, only nine cases were clearly in the position of the case containing details about the domestic conditions of the parties, including the process of changing religions and the forms of violence that occurred. The other 71 cases told more of incompatibility but did not specify the form of incompatibility, and there was also no information about the forms of violence that occurred or the process of changing religions.

The researcher's use of some materials from online newspaper articles and court decisions as research material was caused by several considerations. First, the issue of a husband and wife's different religions is a sensitive and taboo subject to discuss with other people (except in trials in front of judges). To do so needs a long process of building trust if the researcher decides to continue this research by collecting empirical data. Divorce is not something that is generally discussed openly in public, except when it occurs among

artists or public figures in Indonesia. The reasons for divorce are often reluctant to be discussed by people, especially when related to one of the parties changing religion or apostasy (murtad).

Second, divorce cases and inheritance cases are court cases that often contain stories of domestic violence. In the judge's verdicts (not all judges' decisions), especially in the case position, traces of the story of violence are visible. Sometimes people who never read the verdicts from feminist legal perspectives or gender studies perspectives will question 'why do the victims of violence not bring their case to the criminal court through a report to the police?'. They are usually worried about the slow process of criminal justice and want to end the marriage relationship immediately.

Third, this research was conducted in 2020, when the pandemic period had just begun and it was difficult to research in the form of interviews with parties outside the researcher's domicile area, considering the travel ban imposed. Not all regions also have good internet facilities for conducting online meetings, especially for judge interviews. Not all courts, even those in the area closest to the researcher, are willing to provide document search or interview services. This happened because, in several courts, both court staff and judges were also affected by COVID-19, so that the courts where they worked were forced to temporarily not provide services to the public.

Fourth, tracing the court's decision is also not an easy job, even though it is carried out using a directory of judges' decisions. Not all of these decisions have a complete file archive. In fact, of the approximately 60 decisions obtained, only 15 decisions have a complete archive. Experience and knowledge related to strategies for reading court decisions are required. It may be added that this research is also not easy for a high school student to do.

Reza Banakar, a professor of socio-legal studies, used text reviews in his research on Ombudsman performance in Sweden. In this study, Banakar used analytical techniques for two documents, a letter of complaint, and a record of how the dispute was processed by the Ombudsman (Banakar 2005). According to Banakar, the analysis of this text provides empirical data related to the case. Departing from what Banakar did, the researchers analyzed the reality experienced by the parties involved in the divorce case because of the apostasy aspect, where the reality lies in the position of the case.

Bettina Lange (2005) in her article also made use of text analysis and behavioral observation as one of the methods in conducting research related to law in action. Lange analyzed documents related to the EU Directive on Integrated Pollution Prevention and Control and then compared them with the results of an analysis of conversational texts from his interviews with several related officials (Lange 2005, pp. 186–87). This paper is of course not as comprehensive as Lange's writing, but the method used by Lange is applied by comparing the facts about the family conditions of the parties narrated in the case position section with the perspectives of the judges narrated in the considerations and decisions section.

Data from the case position section in court decisions and online newspaper news reflect the realities of legal perception, legal anthropology, and gender studies. However, of course, the depth of this reality can still be debated, especially in anthropological studies, for example, which are very detailed in extracting and narrating data.

In the considerations and decisions of judges in court decision documents, analysis is carried out on the perspective used by judges (Irianto 2020). For example, related to the judge's perspective on the position of women (wife) and men (husband), the judge's perspective on religious freedom in the family, and whether the judge is sufficiently able to recognize the signs of domestic violence from the information presented.

This aspect of reality is also explored in online newspaper news. Of course, there will be questions, regarding the confirmation of the news, and how to avoid clickbait used by the media, or even hoaxes.

## 4. Results

The rights of religious freedom in Indonesia, although protected by the UDHR, The Constitution of The Republic of Indonesia, and by the Act of the Republic of Indonesia of Protection of Human Rights No. 39/1999, still face various challenges in implementation. The choice of religion in some families is highly influenced and determined by the authority in the family. The efforts of women or children in the family to exercise their right to freedom of religion often clash with the power relations and interests of the authorities in the family. Clashes between rights and restrictions imposed within the family often lead to forms of domestic violence.

Domestic violence is simply defined as a series of forms of the use of violence or threats of violence ranging from psychological, emotional, physical, sexual, and neglect. The purpose of this violence is to control spouses or children or other family members who live or are within the scope of the household (Wulandari 2020).

That definition of domestic violence is similar to the domestic violence definition based on the Act of The Republic of Indonesia No. 23/2004 of the Elimination of Domestic Violence, in Article 1 verse (1). That Article states that domestic violence is any act against someone, especially women, which causes suffering and grievances physically, sexually, psychologically, and/or financially, including threats to commit acts, coercion, or illegal deprivation of liberty within the scope of the household.

Why can domestic violence occur, even when the matter is related to issues of religiosity? In situations where there is an unbalanced power relationship in the family and the authorities feel the need to state their actions by using violence, then this violence is a way to strengthen the control of the authority holder over other family members. This control measure using violence also occurs when a family member is deemed to have deviated from the religious teachings of the family. The violence that is deemed necessary to be used to solve problems and deviations in the family, in the culture of society is often not seen as violence, but rather, as part of the way of education (Wulandari 2020, p. 217).

How violence is seen as a tool to discipline family members could be seen, for example, when researchers conducted interviews with several religious leaders and public officials in East Nusa Tenggara in the context of other research conducted between 2017 and 2019 (before the pandemic). In this study, one public official and one regional leader (both male) stated that "there is no violence against children and women in this area. If you beat your mother (wife) and child, you are disciplining the mother and child".

Two pastors in different interviews also complained that "It is very difficult to change perspectives regarding violence in the family is violence and not education". This statement was also reiterated by women who work as victims' companions in institutions that provide services for victims of violence.

The assumption that acts of violence are part of the education of family members applies to some community cultural practices. Thus, when violence occurs, people outside the family do not dare to intervene to assist victims who experience violence, because the community thinks that there is an effort to discipline the victim. Even society thinks that the victim commits the violation. Thus, the victim deserves to be punished as the way to be disciplined.

### 4.1. State Perspective on Religious Freedom (in Indonesia Context)

Before discussing some cases of domestic violence that can illustrate how religious freedom in the family is limited by the issue of power relations, it is important to look first at the State's perspective on religious freedom. In the Constitution of the Republic of Indonesia, freedom of religion is stipulated as a basic right, which is protected by the State. Two articles specifically mention the right to freedom of religion, namely Article 28 E and Article 29.

Article 28E Section (1) stipulates that every Indonesian person has the right to have a religion and practice belief according to the religion, to choose education and teaching,

to choose a job, to choose citizenship, to choose a place to live in the territory of the country and leave it, and still have the right to return.

Then, Article 29 states that every person in Indonesia has the right to have religion or belief and to practice the religion or belief. This Article specifically regulates religious life in Indonesia. Article 29 in Section (2) stipulates that the State guarantees the freedom of every resident to embrace his/her religion and to worship according to his/her religion and belief.

The state perspective contained in the constitution is related to the right to freedom of religion and worship, which is then also translated into legislation. One of the Acts that specifically regulates the protection of human rights is the Law of the Republic of Indonesia No. 39/1999 on Human Rights. In Article 4, it is stated that every Indonesian citizen has:

> " ... the right to life, the right not to be tortured, the right to personal freedom, thought and conscience, the right to religion, the right not to be enslaved, the right to be recognized as a person and equality before the law, and the right not to be prosecuted based on the law which applies retroactively are human rights that cannot be reduced under any circumstances and by anyone ... "

Based on the legislation, the rights to freedom of religion and worship of Indonesian citizens are protected. However, what about in practice, especially in the sphere of the family as the smallest unit in society?

*4.2. Domestic Violence and Divorce Case Report: Something Hidden behind That Numbers*

Based on the Annual Notes of the National Commission on the Elimination of Violence Against Women in 2020, it is known that there were 11,105 cases of domestic violence reported. These domestic violence cases are caused by complex factors. One of the causes is the limitation of religious freedom in the family. The practice of religious freedom in the family does not only have the potential to experience restrictions and violence if these restrictions are met with resistance from parties trying to access their freedom. However, efforts by either party to access the right to freedom of religion and the right to worship can open up opportunities for divorce.

One example that emphasizes the link between restrictions on religious freedom, domestic violence, and divorce as a solution is the data from the religious court on divorce because one party has changed beliefs. In the religious court in Semarang, the number of divorce cases based on the 'murtad' factor reached 40 cases in 2019 (Adi 2019).

During 2019–2020, LBH APIK Jakarta (Jakarta Women Legal Aid Organization) received 125 complaints of domestic violence cases (Denita 2018). Of these, most of the acts of domestic violence occurred because of economic problems or the presence of third parties. However, there were also cases of domestic violence that occurred because a family member had changed religions. LBH APIK Jakarta does not mention the exact number of cases[1], but they show the publication of one case of domestic violence against a young woman who had converted her faith (LBH APIK Jakarta 2019).

One domestic violence case was experienced by a woman who decided to convert her religion. The perpetrators of the violence were the woman's parents and relatives. The forms of violence that were carried out ranged from intimidation to physical violence. Women Legal Aid Organization called LBH APIK, which provided the legal aid for that woman in Jakarta (the capital of Indonesia), and also had to face a group of people brought by the victim's family because the legal aid provider was accused of hiding the victim. That group tried to break into the office of LBH APIK Jakarta (Danu 2020a). The local police even intervened (Danu 2020b).

Some media reported a divorce that occurred because of apostasy aspects. The concept of apostasy here is that one party (husband or wife) changes beliefs. The converting process of belief creates conflict between husband and wife. Then, one of the parties submits an application for divorce, either through the religious court or the local district court.

In the Indonesian context, the party filing for divorce through the religious court is a couple who is married based on the provisions of the Islamic religion and who registered

their marriage at the local Office of Religious Affairs. Meanwhile, those who registered their marriage at the Civil and Population Registry Office are those who are married according to the provisions of other religions outside of Islam. So, the court that has the authority to examine, hear, and decide civil cases related to the affairs of Muslims is the religious court. Meanwhile, the court which has the authority to examine, hear and decide cases related to the affairs of non-Muslims is the district court. It is also important to understand that criminal cases that occur against or are committed by Muslim and non-Muslim residents are fully under the authority of the district court to examine, hear, and decide the case.

This regulation regarding the authority of the court is regulated in Act no. 48 of 2009 concerning judicial power. Based on Article 25 (Act No 48 of 2009), the highest judicial power in Indonesia is under the Supreme Court. Then, under the Supreme Court is the High Court, which is located at the provincial level. Below the high court are general courts for general criminal and civil cases, as well as religious courts, military courts, and state administrative courts; all of these courts are at the district or city level. The religious court (Paragraphs 1 and 3) has the authority to examine, judge, decide, and settle cases between people who are Muslim under the provisions of the legislation.

For private law cases (marriage, divorce, inheritance), some Indonesian Muslims settle their private law cases at the religious court or 'Pengadilan Agama' at the district or city level. When the parties feel that the religious court's decision is not fair enough, they have the right to appeal to the High Religious Court. Those High Religious Courts are located in the capital of the province. If there are still problems, then the case will be raised to the cassation level, which is submitted to the Supreme Court.

Some Indonesians who are not Muslim go to the general court to settle their private law cases (marriage, divorce, inheritance). The first level general court in Indonesia is 'Pengadilan Negeri' or the district court at the district or city level. When the parties feel that the district court's verdict is not fair enough, they have the right to appeal to the High Court, which is located in the capital of the province. If there are still problems, then the case may be sent to the cassation level, which is submitted to the Supreme Court.

Why do Indonesian people need to go to a different court according to their religion to apply for a divorce? Apart from the fact that Law No. 48/2009 stipulates that the private affairs of Muslim Indonesian citizens are under the authority of the religious court to regulate it, also because of the rules that apply to marriage in Indonesia. Marriage in Indonesia is regulated in Law No. 1 of 1974 concerning marriage, especially in Article 2, which states that marriage is only declared valid and recognized by the State if it fulfills two conditions. First, the marriage must be carried out according to the religious law as regulated in Paragraph (1). Second, it must be registered according to applicable laws and regulations or according to State law as regulated in Paragraph (2) of this Article.

It is not stated that the couple who are getting married must have the same beliefs. However, the implementers of these regulations, starting from the village level up to the State and religious leaders, interpreted the regulations to mean that a marriage can only be said to be valid if it is carried out according to the laws of each religion. The parties who have the authority to legalize marriage according to the laws of each religion in effect in Indonesia have various meanings related to the permissibility of couples of different religions to marry. However, the widely used interpretation is that marriages conducted by couples of different religions are considered invalid or as far as possible are not carried out or not given permission by the authorities.

The arrangement of how this marriage should be carried out by taking into account the similarity of beliefs held by the husband and wife has several consequences. First, marriages in Indonesia are difficult to carry out between followers of different faiths. Second, in circumstances where a couple who have different religions from each other are still getting married, one of them will convert to the religion of the partner. It often happens that in societal cultures where patriarchal values are still very strong, religious conversion is carried out by women who follow the religion of their future husbands. Third, over the

course of the marriage itself, it is not impossible for parties who convert to return to their original religion. This will create conflict within the marriage.

In several religious courts in Indonesia, it was noted that the cause of divorce was due to a factor of religious conversion which eventually led to a dispute or conflict between husband and wife. For example, at the Semarang City Religious Court (Pengadilan Agama Kota Semarang), it was noted that there was an increase in the divorce rate due to the husband's conversion to religion in 2019. This was conveyed by the Junior Clerk of the Semarang City Religious Court. According to him, that number has doubled compared to 2018. The Semarang City Religious Court only accepted 19 cases of divorce due to apostasy in 2018. However, between January and December 2019 the Semarang City Religious Court received 38 divorce cases for apostasy (Hardianto 2020).

In September 2019, according to the Junior Clerk of the Semarang City Religious Court, there were eight divorce cases due to conversion. Based on the Court's records, this number is higher than in other months in the same year. The overall divorce rate alone in 2019 reached 3403 cases submitted to the Semarang City Religious Court (Chandra 2019).

The increase of divorce cases also occurred in the Madura and Blitar Religious Courts in 2019–2020. In 2019 the divorce cases filed in the Madura Religious Court were 3947 cases, which are 15 cases that occurred due to changes in the religious beliefs of one party. In the religious court of Blitar, the divorce case filed reached 4365. From those cases, 10 cases were caused by polygamy and converting of religion (Basri 2019; Erliana 2020).

Issues of divorce due to bickering and apostasy have also appeared in other district court statistics. For example, in the records of the Cibinong District Court, Bogor Regency, the total divorce rate in 2019 was 3880 cases. The main cause of divorce was economic factors. However, there were also five divorce cases where one party had converted or apostatized (Metropolitan 2019).

According to media reports, which were traced using the keywords: domestic violence, divorce, and religious differences, there were various factors causing divorce. The factors that cause divorce include economic factors, infidelity factors, and also domestic violence factors. In divorce cases based on the occurrence of domestic violence, one of the drivers of this domestic violence act is the issue of religious differences and changes in the beliefs of a husband and wife. It is interesting that in cases of divorce due to economic reasons, the party filing a divorce suit is a woman or wife (Permana 2020). On the other hand, in some divorce cases caused by the husband or wife convert to other religions, the claims were mostly filed by the men or husbands Adi 2019)

Both Wulandari (2020) and Nafi (2020) articles related to domestic violence and divorce do not mention the violence that occurs related to being forced to believe a certain religion. However, it turns out that in divorce cases submitted to the district court based on the court's statistics report, there are some divorce cases caused by apostasy. This is also confirmed by the court decisions examined in the next section.

The forms and types of domestic violence that were carried out varied, starting from verbal abuse, psychological violence, neglecting and financial violence, and the worst was physical violence. In some cases, husbands or parents impose strict controls and restrictions on the freedoms of those committing to convert to a religion. Perpetrators of the violence could be parents, husbands, siblings, and extended family. Forcing someone to embrace a certain religion is not included in the form of violence that is often carried out in the domestic sphere.

The act of forcing people to embrace a certain religion, even if it is their partner, is an act that is contrary to the principle of religious freedom which is regulated both in the Constitution and international principles on human rights. In some cases, the act of forcing people to embrace a certain religion is accompanied by psychological and even physical violence. For example, in the case handled by LBH APIK Jakarta, where the woman was subjected to violence and attempts to deprive her family of independence for changing religions.

In the next section of this paper, several cases of household violence experienced by someone because he changed his belief are described. These cases were extracted from media reports. The selection of these cases was based on the discovery of three main keywords, namely domestic violence, change of belief, and divorce.

*4.3. Digging the Case of Domestic Violence from Online News to Discover the True Practice of Religious Freedom*

In cases obtained through online news searches, forms of violence perpetrated by perpetrators of domestic violence against their victims are described. Then, the reasons for perpetrators of domestic violence, including reasons related to restrictions on freedom of religion and worship according to the victim's will. This is different from the content of court decisions which mostly only mention the words: conversion of faith, apostasy, causing disputes or incompatibility or quarrels between the parties, and being rude. Several decisions also did not mention the gender identities of the plaintiffs and defendants so that it was rather difficult for the author to analyze the power relation aspects of the case.

On the other hand, in case tracing through online news, information about the marital status of the parties (perpetrator and victim) is not always obtained, and whether they carried out a marriage according to certain religious laws. Then, online news often puts clickbait in the form of a bombastic title, even though the content is not as written in the title. Not only that, but several issues also need to be rechecked to obtain comprehensive news.

The first example of domestic violence that occurs because a person changes beliefs or chooses a different belief from his family is the case experienced by F, a celebrity. Initially, F divorced her husband. In this case, the celebrity from the Instagram platform was divorced by her husband because she was deemed not to have fulfilled the obligations regulated in religion according to her husband's perception. For example, not obeying the husband's orders, and resisting when her husband reminded her about something. It was a clue that the violence was verbal and psychological abuse by the (ex) husband (Muhammad 2019). After the divorce, however, there was a status post from the celebrity, who expressed relief. The woman later converted her religion after divorcing. This caused her father, who is a public figure in the community, to commit verbal violence against this woman. The woman then had isolated herself from the family, society, and media. However, there was then a rapprochement between the woman and her family (Muhammad 2019).

The second example is the domestic violence experienced by a young woman called D. In the case of D, which occurred in February 2020, the perpetrator of violence was her nuclear and extended families. The family objected to D having a relationship with a man of a different religion. D then ran to the city where her boyfriend lived, even converting her religion and making plans to marry her boyfriend.

D's family was very angry and tried to persecute D and her boyfriend/husband. Not only that, but D's family also intimidated D's husband's parents. D then cried for help. She contacted a legal aid agency to provide legal assistance. The legal aid office also sent her to the safe house.[2]

However, the family then broke into the office of the legal aid organization. It did not stop. D's family then reported the case to the police, alleging that D's girlfriend hid D with the help of the legal aid agency. As a result, several police officers then came to the office of the legal aid agency to conduct an investigation[3]. According to the local police, their presence at the legal aid agency's office was for the purpose of fulfilling the request of D's parents who were looking for their daughter who was hidden by her boyfriend. D was only willing to be met at the legal aid agency's office. However, D was not willing to be met by her parents. This caused the anger of D's parents, such that D's extended family came to the office. The police were trying to secure the situation at the office, according to the police.[4]

Thus, in D's case, there was violence perpetrated by D's parents, not only to D as an individual but also to other people. However, the violence was also carried out by D's parents to third parties who helped their daughter. D's family intimidated D's boyfriend/fiancée

and his parents. Then, they also used verbal and psychological violence, threats of physical violence, and broke into the office of the legal aid agency that protected D.

D experienced intimidation, verbal and psychological violence from her family. Her parents also limited D's access to her rights to choose her own belief, and to marry someone that she loves. D's choice in resolving the prolonged conflict between herself and the family was to involve a third party and enter legal channels. In this case, D initially chose to do adjudication (Patresia 2020).

At the beginning of the case, D tried to avoid the violence. However, over time, because the family had chosen to intimidate and carry out coercive action, then D chose to break off her relationship with her parents. Then, D chose the legal option. Unfortunately, state law cannot fully protect D's right to choose her belief, or even to be married to her boyfriend. Law enforcement officials are also very careful in handling cases related to religious conflicts because religious matters always are sensitive matters for most of the Indonesian people (Adam 2019).

The case experienced by D also happened to N. In N's case, her mother prohibited N from having a relationship with her boyfriend who had a different religion. Meanwhile, N and her boyfriend secretly had a relationship since 2018. However, then N's mother knew about her daughter's boyfriend. The climax of the conflict between N and her mother occurred when N secretly changed religions in November 2019. Threats, curses, even prayers for N to die were made by the mother. On several occasions, N's mother also threatened to divorce N's stepfather (Patresia 2020).

In this case, N experienced verbal and psychological abuse by the mother. Her mother demanded that N not only should obey her parents but also continue to embrace the same beliefs as N's mother's religion. Unlike the case with D, who sought help from a legal aid agency, N persisted in not reporting this case. She chose to avoid her mother to reduce the conflict (Patresia 2020).

The fourth case example is case R. The woman was a widow who had two children. R, after working with a man of a different religion, married and then converted. R had made a video stating that she had changed her faith because of her own will. As for R's extended family and residents of her village, they were very angry and tried to make R return to her original religion (Sodikin 2020). R refused. The family then took R's two children and forbade them to meet R.[5]

R experienced verbal and psychological violence against her. The family were also violent towards R's children. The extended family of R attempted to cut off the relationship between R and her two biological children. The family took R's children and took them back to the village (Armando 2019).

The fifth case was experienced by an Indonesian television film actress, with the initials M. This woman was dating and even later married a man of different beliefs. It is not certain whether this woman later changed her belief. However, the actress's family refused to allow the actress's husband to come to meet the family. Even when the actress's parents were interviewed, the parents issued a statement rejecting their son-in-law and did not recognize the man as a son-in-law.[6]

In M's case, although there was no change in belief in M, the parents refused to acknowledge the relationship between their child and the man of the different religions. The family even blacklisted the man. According to M's father, if the man dared to come to their house, he would face rejection from the family or be considered non-existent (Jonata 2014).

The sixth case involved a man with the initial S who converted. The S family is prominent in Indonesia. Initially, when they heard that S changed his faith, his family's reaction was to not believe him. Some of S's siblings then refused to meet S. Some were angry and said harsh words. However, S then still tried to meet his family, and re-establish good relations. Finally, the S family could accept S again (Husna et al. 2020).

S's experience in dealing with forms of verbal violence on the part of his family could then be handled properly. S could be accepted back by his family even though it was not

easy. It is interesting to see that in the previous five cases, the women who had changed their beliefs acquired forms of verbal and psychological domestic violence that might even be quite intense and difficult to stop. In the case of S, the change initially caused conflict but did not drag on. Is there a problem of constructing a viewpoint in society that is more pro-men than women in the 'right' to make decisions? Or perhaps was it the more open-ended value construction within the S family?

In the seventh case, a man experienced domestic violence in the form of psychological pressure exerted by his girlfriend's extended family. The psychological pressure even led the man to commit suicide in the end. A man named B reunited with his old lover named K. B and K then decided to get married secretly. However, when he was about to get married, K asked B to change his belief first. The conversion was a term from K's family so that B may marry K. B agreed because he was afraid that the residents would also accuse B and K of committing adultery. Before getting married, B then performed a ritual to convert his religion. However, on the night after the marriage, B then asked permission to go out from K's house (Chandra 2020).

B never returned home. In the morning, some people found B's body hanging from a tree by the side of the highway. K's family questioned the cause of B's death. However, in the end, B was declared by the local police office as having committed suicide.

In this case, B experienced a form of psychological violence on the part of K's extended family to marry and change religions. On the other hand, B also experienced pressure from his own family to continue to embrace his native religion. When this pressure could not be managed properly by B, he eventually committed suicide (Chandra 2020).

In the cases extracted from online media, it appears that the right to religion, in the Indonesian context, is not a human right that is entirely within one's authority to decide the use of that right. The choices of religion in Indonesia are ruled by the parents, extended family, or the community. The family uses various methods to strengthen its authority over individuals related to the exercise of the religious rights. Based on some cases from the media, parties who try to access their rights related to religious practices, especially in changing beliefs, tend to experience rejection from their families in the form of violence, ranging from verbal to physical violence.

In addition to investigating cases of domestic violence related to changes in religion or belief by a person, this article also investigates and analyzes several court decisions. The court decisions chosen are those related to divorce which was triggered by domestic violence, and there were indications of a change in belief or religion on the part of one of the parties.

### 4.4. Court Decisions, Judges Perceptions

This section describes the analysis of court decisions related to domestic violence cases that lead to divorce. The reasons for the occurrence of domestic violence, among others, were because one of the parties changed religions, or returned to the original religion that they had before they get married.

### 4.4.1. Religious Court of Pasuruan Verdict No. 970/Pdt.G/2009/PA

In this case, the plaintiff and defendant had been married for four years but had no children. At the beginning of the marriage, the plaintiff and defendant's family conditions were harmonious. They lived in one house.

Then, in 1998, in their fourth year of marriage, the plaintiff filed for divorce. The reason was that the defendant was found secretly entering one of the places of worship. The plaintiff suspected that the defendant had converted, and returned to his old belief before marriage. When asked by the plaintiff, the defendant did not admit that he entered the house of worship.

Initially, the plaintiff expressed their objection and quarrel with the defendant. However, then the defendant left the house where he and the plaintiff were living together. When the plaintiff came home from work, she found that the defendant was gone.

The address of the defendant was not known because he lived in different places. Finally, the defendant was found to have lived with his friend. Meanwhile, the plaintiff moved to the plaintiff's parents' house.

During the trial process, the defendant did not appear in court or send a representative. The panel of judges tried to advise the plaintiff and made mediation efforts. However, because the defendant did not know where he lived and was not present during the mediation process or at the trial, the mediation effort could not be carried out. The plaintiff and the witnesses also stated that the discrepancies and disputes could no longer be reconciled. Mainly for the reason that the defendant had apostatized by returning to his old belief before he married the plaintiff. Thus, the panel of judges granted the divorce suit filed by the plaintiff.

### 4.4.2. The Religious Court of Wates Verdict No. 57/Pdt.G/2014/PA.Wt

In this case, the plaintiff, who was 29 years old, and the defendant, who was 30 years old, had been married for two years but were not blessed with children. The right of the petitioner was to file a request for divorce to the religious court because the respondent was considered to have committed an apostate act.

The defendant returned to his/her origin religion. When they married, the defendant had agreed to follow the plaintiff's religion. The plaintiff and the witnesses then explained to the panel of judges in the trial that efforts had been made to persuade the defendant to convert to the same religion as his/her partner. However, the defendant refused. The defendant then left the house without permission and returned to the defendant's parents' house.

The panel of judges then decided to grant the divorce petition on the basis that first, the defendant had apostatized or left their religion when they were married. Second, there had been a dispute that could not be reconciled, as stated by the plaintiff. Third, the defendant had left the house where he lived with his/her partner without the partner's permission. The defendant was not ever-present at the trial, even though the court had sent a summons to him/her several times.

### 4.4.3. The Religious Court of Muara Bulian Verdict No. 256/Pdt.G/2012/PA.Mbl

In this case, the plaintiff is the wife, and the defendant is the husband. The plaintiff and defendant had legally married at the local Office of Religious Affairs. The age difference between the defendant and plaintiff was quite large; the plaintiff was only 33 years old, while the defendant was 62 years old. At the start of their marriage, things went well. However, after the age of marriage reached its second year and produced one child, quarrels began to occur.

The cause of the dispute was partly because the defendant often beat the plaintiff. The plaintiff was also upset because the defendant was often caught teasing other women. The defendant was also not happy that the plaintiff was carrying out his religious obligations.

The plaintiff also found that the defendant did not perform worship according to their religion when they were married. However, the defendant returned to his old religion. According to the plaintiff, this was the cause of the dispute, besides the defendant's habit of physically abusing his wife. The defendant's attitude and actions caused the plaintiff to no longer be able to live with the defendant. Thus, the plaintiff filed a divorce application.

The judge then granted the plaintiff's request for divorce. The consideration of the panel of judges in deciding the divorce was because the defendant had apostatized and then committed violence against the plaintiff. Finally, the decision for divorce was also handed down by the panel of judges because the defendant could not be heard, because he was never present at the trial.

### 4.4.4. The District Court of Purwodadi Decision No. 5/Pdt.G/2018/PN Pwd

This case was resolved at the Purwodadi District Court because the divorced parties married according to the Christian religious procedure. They married in 2007. Throughout the marriage, they were blessed with two children.

In 2009, the plaintiff then decided to return to their original religion. The defendant, who was the wife's party, was invited to change religions. However, the defendant refused to move and remained in her original religion.

Since the plaintiff decided to change religion, there were frequent quarrels and arguments between plaintiff and defendant. According to the plaintiff, this was because the defendant did not want to embrace the same religion as the plaintiff. However, in the defendant's answer and witness testimony, it was found that the plaintiff had committed violence and there were suspicions of infidelity.

The witnesses stated that the plaintiff initially had a different religion from the defendant. However, when the plaintiff was married, they decided to embrace the same religion as the defendant. After two years of marriage, the plaintiff finally returned to his original religion. Although the defendant was not willing to embrace the same religion as the plaintiff, she allowed her two children to be educated according to the plaintiff's religion. The defendant also stated that he had no problem with the change in religion. However, it was the plaintiff who was looking for ways to divorce.

This case was resolved in the district court and not in the religious court because even though the plaintiff was of a certain religion, he had married the defendant according to Christian religious procedures; thus, the divorce case was submitted to the district court, not to the religious court.

The panel of judges then granted the divorce request. The judge considered that both plaintiff and defendant could not be reconciled even though the defendant did not wish to divorce. According to the panel of judges, the incompatibility between the plaintiff and defendant could not be resolved. With the consideration of breaking the chain of violence and dispute, the plaintiff's request for divorce was granted by the panel of judges.

### 4.4.5. The District Court of Medan Verdict No. 102/Pdt.G/2020/PN Mdn

In this divorce case, the plaintiff and defendant had the same religious background. These two married in 1998 in the church based on Christian values. However, in 2010, the defendant started joining the Charismatic sect. This caused a quarrel between plaintiff and defendant. Finally, they slept in separate beds.

In 2011, when the plaintiff's brother died, the plaintiff and defendant made peace at the request of the defendant. The plaintiff's family provided a condition that defendant must leave the Charismatic sect and return to worship as at the beginning of the plaintiff and defendant's marriage. The defendant agreed. Then, the plaintiff and defendant returned to live together.

It turned out that the defendant was again carrying out worship with the Charismatic sect ritual. This raised the conflict again with the plaintiff. Then, in seeking peace, the plaintiff decided to embrace another religion that was different from the plaintiff's original religion. There was an urge to divorce the defendant because of the ongoing dispute.

The defendant gave different information. According to the defendant, even though they worshiped following the rituals of the Charismatic sect, the defendant still served the plaintiff well. When they did not live in the same house, it was because the plaintiff had not repaired their house and the house was dangerous for the children. The defendant also explained that since the plaintiff did not live with the defendant, he neglected the children, and even experienced economic violence in the form of not providing support for the family.

The panel of judges then decided to grant the divorce application submitted by the plaintiff. The reason for the judges was that there had been ongoing and irreconcilable disputes between plaintiff and defendant. Then, the plaintiff had also changed beliefs,

so that it was impossible for the marriage to continue according to the perspective of the plaintiff's belief.

4.4.6. The Religious Court of Binjai Verdict No. 21/Pdt.G/2019/PA Bnj

In this divorce case submitted to the Binjai District Court, the plaintiff and defendant were married based on Christian values in the church. The plaintiff, when she got married, was a maiden. However, the defendant was a widower with two teenage children. The plaintiff and defendant's marriage happened in 2010.

While bound in marriage, the plaintiff and defendant were blessed with two children. However, the plaintiff was often rude and violent towards his children. The defendant and plaintiff also often quarreled and fought. The defendant also committed acts of violence against the plaintiff. The violence was carried out because the plaintiff decided to return to her original religion before she was married. Her parents-in-law also made threats to shoot the plaintiff. Based on the consideration that both the defendant and his family had committed acts of violence, the plaintiff filed a divorce application.

Apart from reasons of violence committed by the defendant, the plaintiff also filed a divorce application because she had returned to her religion, which was different from the defendant's religion. According to the plaintiff's argument, in her religion, it is prohibited to marry men of different religions. Violation of that rule would be considered sinful. On the other hand, according to the defendant's religious values, interfaith marriage is also prohibited, according to plaintiff's statement. Strangely, this argument that the prohibition of marriage with different religions did not come from the values of the defendant's religion, but was based on the plaintiff's interpretation of Article 2 of the Act of Marriage No.1 of 1974.

Article 2 Paragraph (1) of Law Number 1 Year 1974 concerning marriage, states: "Marriage is legal if it is carried out according to the law of each religion and belief". The elucidation of this Article includes a statement "that there is no marriage outside the law of his religion and belief". Thus, the sentence "there is no marriage outside the law of their religion and belief" is interpreted by the plaintiff as: "individuals who have different religions and beliefs cannot possibly marry" even though the legislators intend that marriage between two people of different religions is possible but it still has to be carried out in the corridor of religious law and beliefs.

However, in this case, the legal product produced by the panel of judges used the term stipulation or order, because the situation faced by the plaintiff and the defendant was deemed irreversible. After all, the plaintiff was a woman who embraced a certain religion, and by her religious teachings it was strictly forbidden to marry a man who had a different religion (according to the interpretation of a particular sect in that religion). If the marriage is still maintained, the woman will sin. Thus, the panel of judges did not want to take the risk of taking part in putting someone as sinful. So, based on the judge's perspectives: nothing should be reconsidered.

In this decision, the panel of judges included several considerations:

a.　Marriage must be based voluntarily to achieve happiness, so if the parties are constantly arguing and cannot be reconciled then divorce is the last solution.

b.　Marriage between women and men who embrace different religions is not possible. Moreover, in the religious teachings of the plaintiff before marriage, it was said that marriage to a man of different religions would cause sin for the woman. Thus, the marriage cannot be continued.

c.　Marriage of the plaintiff and defendant is carried out with or according to the ordinances of the Christian Church, thus the authority to adjudicate cases rests in the hands of the district court and not in the jurisdiction of the religious courts under the regulations regarding the jurisdiction of court institutions according to Law No. 48/2009 concerning judicial power. Based on these considerations, the plaintiff's request for divorce was granted by the panel of judges.

### 4.4.7. The Religious Court of Wates Verdict No. 302/Pdt.G/2014/PA.Wt

In this verdict, especially in the case position, it is narrated that a male civil servant was married to a female civil servant. They both had different religions. However, before marrying the woman, the man converted to the woman's religion, as required by the family of the woman.

However, in the process of marriage, there were forms of violence between the husband and wife. The husband then filed for divorce from his wife.

In the trial process, it was revealed that the husband as the applicant turned out to have returned to the practice of his native religion, causing a dispute between the husband and wife. The judge, considering the case based on the Marriage Law and the Principles in the Compilation of Islamic Law, then granted the divorce but on the grounds of apostasy.

### 4.4.8. The Religious Court of Central Jakarta Verdict No. 96/Pdt.G/2013/PA.JP

In this decision, the plaintiff applied for divorce because the plaintiff returned to her original religion before marriage. The plaintiff also knew that the defendant had another wife. Based on the condition of the plaintiff returning to his original religion, the defendant often commits violence against the plaintiff. Another reason for the divorce was that the defendant did not provide financial support for the plaintiff. Based on the Act on the Elimination of Domestic Violence in Indonesia, economic or financial neglect is also part of domestic violence. However, it is interesting that the judge decided this case by granting the plaintiff's request not based on economic violence. However, because the defendant is considered an apostate.

### 4.4.9. The Religious Court of Kaimana Verdict No. 7/Pdt.G/2021/PA.Kmn

In this decision, the applicant (the husband) filed a divorce suit with the religious court because the wife had turned back to her old belief. At the time of marriage, the wife had agreed to embrace the same religion as her husband. However, the wife turned out to still practice the teachings of her old religion.

Unfortunately, this decision does not contain complete file attachments. Only a summary of the decision is related to the names of the parties, the panel of judges, briefly the case, and the divorce decision.

In the decisions of the judges of the Supreme Court described in this section, it can be found that in the divorce cases that were submitted to the court, when the judges dug deeper, there were several reasons.

First, there was domestic violence in various forms, starting from economic neglect, neglect, verbal violence, and physical violence.

Second, the parties stated that there were restrictions on religious freedom. It is good to choose the same religion in their marriage, because of the interpretation of Law No. 1 of 1974 concerning marriage, that marriage must refer to the law of their respective religion and state law. This rule then has been interpreted by the authorities to mean that the couple should have the same religion. This restriction on religious freedom can be a reason for domestic violence and divorce in the examples of cases presented in this section.

Based on those decisions, which also described that if the party who changes religion or belief is the husband, then the wife can file for divorce without any record of domestic violence in the judge's consideration. The wife's reason for filing for divorce is because the husband is an apostate, so they can no longer live as husband and wife legally according to religious law (and this is also approved by the judge in his consideration).

However, if the wife was guilty of the apostasy (changing belief or returning back to the old religion), then in the divorce lawsuit there must be a complaint that there has been an act of violence (economic neglect, harsh words from the husband to the act of hitting or molesting the wife). This shows that in the cultural construction, women as wives are under the authority of their husbands, and wives must submit to their husbands. To subdue the wife, the husband uses various ways.

This cultural construction where the husband acts as the full authority over his wife and children and has some privilege (including hitting) is a construction of a patriarchal culture. In the context of patriarchal culture, men are the holders of power. Women are implementers (Wadud 2005; Nurbayanti 2020). So, as the holder of authority, men are considered by society with a patriarchal culture as having the privilege to regulate, direct, and control their family members (who are considered subordinates) to obey the rules made by the authority holder. It includes the authority to exercise control over the rights of religious freedom of family members.

## 5. Discussion

Based on the results of online news searches and court decisions, it is found that the position of women in the family is still influenced by patriarchal values. Often, women are also seen as not having the authority to make decisions, even those concerning themselves. This includes embracing a belief or religion that is different from her family or choosing a spouse. In the cases faced by F, D, M, N, and R, because they are all women, the possibility of making decisions becomes more challenging because of cultural construction issues, especially if the cultural construction is still strongly influenced by patriarchal values.

However, then, is the domestic violence that occurs, triggered by a change in one's belief, only happening to women because of power relation issues between genders? Hence, the problem of changing religion in the family is quite complex because it is influenced by several things. First, the dominant religious factor in the family. Second, the construction of relations between genders in the family—including whether or not patriarchal values are dominant in the family. Third, the pattern of communication between the authorities in the family and other family members. Fourth, the condition of the individual who decides to change the belief themself.

The condition of society also affects how a case is about limiting religious freedom in the family. In a society where social cohesion is still strong, the family will certainly try to prevent family members who have the intention to change religions to do so, or if this happens, efforts will be made to return the person to his original faith. Feeling worried that what people say will be wrong one reason is also why parents or other authorities in the family commit violence against other family members who decide to change religions or embrace a different religion.

This aspect of social cohesion is also one of the challenges faced by law enforcement officials, ranging from police to judges, especially when these law enforcement officials are dealing with cases of domestic violence in which there are issues of restricting religious freedom. The issue of religion and belief has always been a sensitive matter in Indonesian society.

The issue of domestic violence is also not something that can be discussed openly, let alone brought to the realm of law because it is considered to be still in the private sphere. In addition, there is an issue of freedom of religion, which is limited by the family itself. So, you can imagine how big the obstacles and challenges faced by law enforcers are when they handle cases of domestic violence related to the issue of restricting religious freedom.

In the case from Binjai District Court, the panel of judges uses the term 'order', or in the Indonesian language known as 'Penetapan', not the decision or 'Putusan' in the Indonesian language. In the context of the courts in Indonesia, the use of the term ruling for a product of the court has the consequence that this case at the first and last level has been decided and that an appeal cannot be made. This is certainly interesting because it means that the verdict is final, and the defendant cannot then submit other forms of legal remedies (Harahap 2016). The use of the word stipulation in a divorce case is usually used when the husband has given the divorce proceedings three times to the wife. Thus, the act of reconciliation cannot be taken, and a legal divorce occurs. The panel of judges only acts to strengthen what has happened or just to make it official. Thus, it is called determination.

There is a special note related to how these divorce cases, which include the issue of domestic violence and the issue of limiting the right to religion and worship, are described

in court decisions. The general courts always put the names of the parties on the verdict. However, the religious court does not. The general courts also revealed aspects of disputes and the occurrence of violence in the form of physical, psychological, and economic neglect. In this aspect of economic neglect, the general courts said in their verdicts as "not providing a living". Meanwhile, in the decision of the religious court, the forms of violence were not disclosed in detail, only referred to as "disagreement", or "irresistible mismatch". Meanwhile, economic neglect is defined as not providing a living or not paying attention to children.

It is interesting to then compare Olsen's writings (Olsen 1995) on state intervention in the private spheres of the family. Particularly, on how the state constructs hierarchies within the family and the 'ideal' model of the family. In the consideration of the panel of judges in court decisions related to divorce cases due to apostasy, the judge always postulated that the family was no longer harmonious because one of the parties changed religion. Thus, the divorce suit was granted.

The state through the courts (and judges) intervened (Olsen 1995) on this religious issue through considerations and decisions that strengthened the perception that a family must adhere to the same religion, even though religion and family are both in the private sphere. The judges give considerations and decisions that strengthen the 'coercion' carried out by one party against the other by using the basis of the interpretation of Article 2 of the Marriage Law and also the principles in the Compilation of Islamic Law.

In the Indonesian context, where religious matters are a sensitive issue and can lead to situations where family conflicts can escalate into horizontal conflicts within society, judges are not willing to take the risk. In addition, judges decisions on cases are based on what they believe, namely the law (which is interpreted), as well as related religious rules (even though judges are representatives of state law).

## 6. Closing Remarks

The right to freedom of religion is a part of human rights. However, in the implementation, this right is not always easily accessible. Religion or belief is indeed part of a person's identity. However, as stated by Steph Lawler (2008), it is significant to remember that identity is constructed not only by the individual but also by his family, environment, or community. A person's identity throughout his life continuously goes through a process of formation, changing both internally and externally, said Lawler. In Indonesian society, religion or belief is strongly attached to a person's identity as the result of the construction of his family and society. The decision to embrace a certain religion or belief cannot be easily taken by individuals, because of the significant role of the family and the society.

Religion or belief is the part of the identity that is closely attached to an individual. Identities are constructing by the person, family, community, and state. However, the decision to change religion and to worship is not easy for Indonesian people.

The challenge in the effort to exercise the right to freedom of religion is clearly reflected in the family. In the family, there is an unwritten rule that family members must follow the beliefs held by the authority in the family, namely the father or husband. Based on the divorce cases analyzed in this paper, it is found that the exercise of this authority can then be wrapped up in acts of domestic violence, and can even lead to divorce. It is interesting that the judges, as representatives of the state, are able to recognize this through several considerations in the decisions of these cases. However, the judge cannot take any action related to the violation of religious freedom because the case submitted was an application for divorce.

Thus, the challenge to accessing religious freedom actually does not only come from the public sphere, such as the prohibition on building houses of worship or the prohibition to carry out worship for minority religious groups. Rather, the biggest challenges related to the protection of religious freedom are in the yards and living rooms of our own homes.

**Funding:** This research received no external funding.

**Institutional Review Board Statement:** Not applicable.

**Informed Consent Statement:** Not applicable.

**Data Availability Statement:** All data used in this article can be accessed openly from online newspaper links, directories of the court verdicts (only for cases that are allowed to be published). The names of the parties listed in the judge's verdicts and newspapers have been disguised. Both data from the directory of court verdicts and online newspapers have some links to access at the reference list.

**Conflicts of Interest:** The author declares no conflict of interest.

## Notes

1  https://www.lbhapik.org/2019/12/siaran-pers-lbh-apik-jakarta-laporan.html (accessed on 10 December 2020).

2  https://magdalene.co/story/jerat-orang-tua-toksik-dan-sulitnya-anak-menentukan-nasib-sendiri (accessed on 30 December 2020).

3  https://news.detik.com/berita/d-4909514/lbh-pembela-perempuan-disambangi-polisi-dan-preman-apa-yang-happened/2 (accessed on 30 December 2020).

4  https://news.detik.com/berita/d-4909558/polisi-tak-ada-penggeledahan-paksa-ke-lbh-apik (accessed on 14 December 2020).

5  https://www.tagar.id/kasus-aceh-cut-fitri-islam-ke-kristen-kenapa-marah (accessed on 14 December 2020).

6  https://www.tribunnews.com/seleb/2014/01/28/asmirandah-dikabarkan-pindah-agama-orangtua-blacklist-jonas-rivanno (accessed on 14 December 2020).

## References

### Primary Sources

International Declaration, National Regulations and Verdicts.
The Universal Declaration of Human Rights.
Republic of Indonesia. Constitution with Amandement.
Republic of Indonesia. The Act No. 1/1974 of Marriage.
Republic of Indonesia. The Act No. 39/1999 of Human Rights.
Republic of Indonesia. The Act No. 23/2004 of the Elimination of Domestic Violence.
Republic of Indonesia. The Act No. 48 of 2009 of the Judicial Power.
District Court of Binjai Verdict No. 21/Pdt.G/2019/PN Bnj.
District Court of Purwodadi Verdict No. 5/Pdt.G/2018/PN Pwd.
District Court of Medan Verdict No. 102/Pdt.G/2020/PN Mdn.
Religious Court of Pasuruan Verdict No. 970/Pdt.G/2009/PA.
Religious Court of Wates Verdict No. 57/Pdt.G/2014/PA.Wt.
Religious Court of Muara Bulian Verdict No. 256/Pdt.G/2012/PA.Mbl.
The Religious Court of Wates Verdict No. 302/Pdt.G/2014/PA.Wt.
The Religious Court of Central Jakarta Verdict No. 96/Pdt.G/2013/PA.JP.
The Religious Court of Kaimana Verdict No. 7/Pdt.G/2021/PA.Km.

### Secondary Sources

Adam, Aulia. 2019. Berita Pindah Agama Cenderung Menyudutkan Penganut Agama Minoritas. Available online: https://tirto.id/berita-pindah-agama-cenderung-menyudutkan-penganut-agama-minoritas-efVZ (accessed on 14 December 2020).
Adi. 2019. Perceraian karena Murtad di Semarang Tahun 2019 Naik 100 Persen. Available online: https://www.kaskus.co.id/thread/5e34508bf0bdb209443f945d/perceraian-karena-murtad-di-semarang-tahun-2019-naik-100persen/ (accessed on 14 December 2020).
Armando, Ade. 2019. Kasus Aceh: Cut Fitri Islam ke Kristen, Kenapa Marah. Available online: https://www.tagar.id/kasus-aceh-cut-fitri-islam-ke-kristen-kenapa-marah (accessed on 14 December 2020).
Banakar, Reza. 2005. Studying Cases Empirically: A Sociological Method for Studying Discrimination Cases in Sweden. In *Reza Banakar and Max Travers, Theory and Method in Socio-Legal Research*. Portland: Hart Publishing and Onati I.I.S.L.
Bartky, Sandra. 2005. Battered Women, Intimidation, and the Law. In *Marilyn Friedman, Women and Citizenship*. New York: Oxford Press.
Basri, Abdul, ed. 2019. 15 Perceraian karena Murtad Setahun PA Terima 3.947 Perkara. November, Available online: https://radarmadura.jawapos.com/read/2019/11/23/167182/15-perceraian-karena-murtad-setahun-pa-terima-3947-perkara (accessed on 14 December 2020).

Chandra, Iswinarno. 2019. Tren Cerai karena Suami Pindah Agama Meningkat Tajam di PA Kota Semarang. Available online: https://jateng.suara.com/read/2019/12/30/134449/tren-cerai-karena-suami-pindah-agama-meningkat-tajam-di-pa-kota-semarang (accessed on 14 December 2020).

Chandra, Iswinarno Senin. 2020. Pindah Agama dan Menikah Diam-diam Diduga Jadi Alasan Pengantin Bunuh Diri? Available online: https://kaltim.suara.com/read/2020/11/02/132345/pindah-agama-dan-menikah-diam-diam-diduga-jadi-alasan-pengantin-bunuh-diri (accessed on 14 December 2020).

Danu, Damarjati. 2020a. LBH Pembela Perempuan Disambangi Polisi dan Preman, Apa yang Terjadi? Available online: https://news.detik.com/berita/d-4909514/lbh-pembela-perempuan-disambangi-polisi-dan-preman-apa-yang-terjadi/2 (accessed on 14 December 2020).

Danu, Damarjati. 2020b. Polisi: Tak Ada Penggeledahan Paksa ke LBH Apik. Available online: https://news.detik.com/berita/d-4909558/polisi-tak-ada-penggeledahan-paksa-ke-lbh-apik (accessed on 14 December 2020).

Denita, Br Matondang. 2018. LBH APIK Terima Aduan 308 KDRT Selama 2017. Available online: https://news.detik.com/berita/d-3867106/lbh-apik-terima-aduan-308-kdrt-selama-2017 (accessed on 14 December 2020).

Erliana, Riady. 2020. Murtad dan Poligami Kini Jadi Tren Penyebab Perceraian di Blitar. Available online: https://news.detik.com/berita-jawa-timur/d-4846907/murtad-dan-poligami-kini-jadi-tren-penyebab-perceraian-di-blitar (accessed on 14 December 2020).

Harahap, Yahya. 2016. *Hukum Acara Perdata: Tentang Gugatan, Persidangan, Penyitaan, Pembuktian, dan Putusan Pengadilan*. Jakarta: Sinar Grafika.

Hardianto, Fariz. 2020. Gara-gara Pindah Agama Puluhan Suami di Semarang Ceraikan Isterinya. Available online: https://www.idntimes.com/news/indonesia/fariz-fardianto/gara-gara-pindah-agama-puluhan-suami-di-semarang-ceraikan-istrinya-nasional/2 (accessed on 14 December 2020).

Husna, Rahmayunita, Dea Dezellynda, and Madya Ratri Senin. 2020. Sempat Menentang, Respons Alyssa Soebandono Tahu Kakaknya Pindah Agama. Available online: https://kalbar.suara.com/read/2020/12/14/155959/sempat-menentang-respons-alyssa-soebandono-tahu-kakaknya-pindah-agama (accessed on 14 December 2020).

Irianto, Sulistyowati. 2020. Teori Hukum Feminis. In *Perempuan dan Anak dalam Hukum dan Persidangan*. Edited by Sulistyowati Irianto and Lidwina Inge Nurtjahyo. Jakarta: Yayasan Obor Indonesia.

Jonata, Willem. 2014. Asmirandah Dikabarkan Pindah Agama, Orangtua Blacklist Jonas Rivanno. Available online: https://www.tribunnews.com/seleb/2014/01/28/asmirandah-dikabarkan-pindah-agama-orangtua-blacklist-jonas-rivanno (accessed on 14 December 2020).

Lange, Bettina. 2005. Researching Discourse and Behaviour as Elements of Law in Action. In *Reza Banakar and Max Travers, Theory and Method in Socio-Legal Research*. Portland: Hart Publishing and Onati I.I.S.L.

Lawler, Steph. 2008. *Identity: Sociological Perspectives*. Cambridge: Polity Press.

LBH APIK Jakarta. 2019. Siaran Pers LBH APIK Jakarta. Available online: https://www.lbhapik.org/2019/12/siaran-pers-lbh-apik-jakarta-laporan.html (accessed on 14 December 2020).

Metropolitan. 2019. 2.504 Istri Gugat Cerai Suami Kere. Available online: https://www.metropolitan.id/2019/10/2-504-istri-gugat-cerai-suami-kere/ (accessed on 14 December 2020).

Muhammad, Ridho. 2019. VIDEO Pengakuan Salmafina Sunan Pindah Agama, Taqy Malik Bongkar Alasan Ceraikan Anak Sunan Kalijaga. Available online: https://pekanbaru.tribunnews.com/2019/07/15/video-pengakuan-salmafina-sunan-pindah-agama-taqy-malik-bongkar-alasan-ceraikan-anak-sunan-kalijaga?page=4 (accessed on 14 December 2020).

Nafi, Tien Handayani. 2020. Perempuan dan Anak dalam Perceraian. In *Perempuan Dan Anak Dalam Hukum Dan Persidangan*. Edited by Sulistyowati Irianto and Lidwina Inge Nurtjahyo. Jakarta: Yayasan Obor Indonesia.

Nurbayanti, Herni Sri. 2020. Konsep-konsep Dasar tentang Gender. In *Perempuan dan Anak Dalam Hukum dan Persidangan*. Edited by Sulistyowati Irianto and Lidwina Inge Nurtjahyo. Jakarta: Yayasan Obor Indonesia.

Olsen, Frances E. 1995. The Myth of State Intervention in The Family. In *Feminist Legal Theory Volume II*. New York: New York University Press.

Patresia, Kirnandita. 2020. Jerat Orang Tua Toksik dan Sulitnya Anak Menentukan Nasib Sendiri. Available online: https://magdalene.co/story/jerat-orang-tua-toksik-dan-sulitnya-anak-menentukan-nasib-sendiri (accessed on 14 December 2020).

Permana, Fuji E. 2020. Banyak Orang Bercerai Saat Pandemi COVID-1. September. Available online: https://republika.co.id/berita/qgkyhn282/banyak-orang-bercerai-saat-pandemi-covid19 (accessed on 14 December 2020).

Sodikin. 2020. Janda 2 Anak di Aceh Pindah Agama, Ini Pengakuan Sang Kakak. Available online: https://www.islampos.com/janda-2-anak-di-aceh-pindah-agama-ini-pengakuan-sang-kakak-193964/ (accessed on 14 December 2020).

Wadud, Amina. 2005. Citizenship and Faith. In *Marilyn Friedman, Women and Citizenship*. New York: Oxford Press.

Wulandari, Widati. 2020. Kekerasan dalam Rumah Tangga. In *Perempuan Dan Anak Dalam Hukum Dan Persidangan*. Edited by Sulistyowati Irianto and Lidwina Inge Nurtjahyo. Jakarta: Yayasan Obor Indonesia.