# Peer review of "The Issue of Rights of Religious Freedom in Some Domestic Violence Cases in Indonesia"

_religions, doi:10.3390/rel12090733_

Round 1

Reviewer 1 Report

I can see the Author worked hard on the manuscript. It is much better now. Only minor issues remain, such as "The man that N loves so much” - a melodramatic statement, not for a research article; just delate "so much" and it will be fine.

Author Response

Dear Sir/Madam,

I have corrected this post according to your input. The parts that have been repaired are colored blue. Thank you very much for your input.

Regards.

Reviewer 2 Report

I find the topic to be an important topic (violence against women due to change of religions). However, I still find the paper lacking important sections.

First, I do not see a clear research question. The authors said that "This paper describes how the practice of freedom of religion and worship..." but fail to talk about how they "describe" systematically. Are they using a certain approach to collect and analyze materials? Are they doing any research? In short, what exactly is the research question?

Then, the paper jumps to results straight from the introduction. There was no literature review.  A manuscript should always build on existing knowledge. Starting from nothing is not appropriate. There are studies out there that look at religious conflicts in families/communities from other countries. The authors should at least review the literature and then design their study approach. 

A method section should precede a result section. What methods were used to collect these data? What research design was used? etc. The authors mentioned in line 731 and onwards; however, the method should be described first so that readers have the context when reading the results. 

Results should be a good summary of findings, rather than a laundry list of things the authors find here and there. The current result section is 11 pages. Instead of describing every source, the authors should use analytical methods to summarize knowledge and generate themes. 

Author Response

Dear Sir/Madam,

Herewith I send the manuscript with some correction according to your input. The parts that have been repaired are colored blue. Thank you very much for your input. Regards.

Reviewer 3 Report

The topic of the paper is timely, compelling, and engaging.. There is a good mix of sources, including court decisions, media sources, and secondary academic sources.

Linguistically and structurally, there are some English errors here and there,  but these can likely be corrected. A more significant structural issue that should be fixed are the abundant short paragraphs of fewer than three sentences. I have noted some specific placed where sequential paragraphs can be combined. There might also be better use of descriptive subheadings to draw out concepts, themes and findings for the reader. 

There were a number of pages highlighted in yellow that seemed as if they might have been insertions following a previous review. Many of them were good additions. However, the lengthy methodology section at the very end was odd--as was the complete absence of a concluding section. 

The methodology section may have been inserted at the recommendation of a previous reviewer. I do not work in a social science filed where these are common. What is normally more common and desirable is to have an introductroy section of no more than 5 pages length (depending on the overall length of the paper) that ends with a strong thesis paragraph that lays out the sections of the paper and perhaps previews some conclusions. That structure shoudl then be carried through the paper with good subheadings and introductions and conclusions to subsections.

If the author is in a field where methodology statements are the norm, then I would recommend moving it up (a short section after the introduction is often scene). Some of the author's more substantive and critical comments in the current methodological section could come after the paper's main analysis (that is, remain in pretty much the same place) as a way of winding to a close with some good critical comments on the merits and limits of the research and possible further avenues. 

But what this article needs most is a good concluding section. Some fields make do with a single paragraph. Mine favors three to five (sometimes more) to review for the reader the article's key findings and what to take away. Also, a nice "bookend" for an effective conclusion is to have the introduction open with brief and compelling summary of a recent case to "problematize" the issue for the reader and to draw them in. A little additional and revision work at the beginning and end of this article, along with tightening up those short paragraphs throughout and relocating and perhaps editing down the methodology section could make this a very good contribution to the literature.

Author Response

(The authors gave the same response as above.)

Round 2

Reviewer 2 Report

The manuscript is in better shape now. 

This manuscript is a resubmission of an earlier submission. The following is a list of the peer review reports and author responses from that submission.

Round 1

Reviewer 1 Report

This article is interesting and has some potential. The usage of documents is good as is the legal basement. The major argument sounds good.

Unfortunately, the article is under researched. Theory is almost none, methodology is minimal, the content is lacking. Superficially, there is much, with documents filling many pages, but when we cut the irrelevant aspects, very little content remains. The Author writes about 7 cases and 6 court cases. Based on this very small variable s/he build the argument and draws conclusions. That's much too little. Academic work is not about googling some examples and adding documents to it to produce a thesis. This would be acceptable, perhaps, for a end-of-term essay at high school, but is it not enough for an article in a journal. Although the argument the Author makes seems reasonable, it is just that: it sounds good. But it is not followed by justification (verification or falsification) based on sufficient evidence. Consequently, it is a sloganeering. The text suffers from too many generalities and too little specific details. E.g. "in the culture of society is often not seen" - this is much too general (what culture? what society? these are too large categories to use here).  

Minor issues:

The documents (UN and Indonesian laws) cover roughly 40% of the article, they should be packed into 2-3 pages. The rest should encompass more empirical content + methodology. 

“media reports, which were traced using key words: domestic violence, divorce, religious differences” - I expect a more sophisticated way of researching than just googling.

“Then online news often puts click bait in the form of a bombastic title even though the content is not as written in the title” - well, this is the nature of the media.

“The man that N loves so much” - a melodramatic statement, not for a research article

An English native speaker's proofreading is really needed, e.g.

"that domestic violence could be happened when women";

"a young woman who has been converted her faith";

"but they show the publications of one cases of domestic violence against a young woman who has been converted her faith. One case of the domestic violence case experienced by a woman who decides to convert her religion." (repetition (and style)

Page: 8 - is it a man or a woman?

“The fourth case example is case R. The woman is a widow who has two children (…)R had made a video stating that he had changed his faith because of his own will (…) tried to get to return to his original religion. R refused (…)R had verbal and psychological violence against her. The extended family of R attempt to cut off the relationship between R and his two biological children. (…, next page: the previous five cases, the women)”

“marriage was happened in 2010”

…..

There is a potential in this article, though s/he would have to work hard on it. The simple way to improve it is to congest (pack) all legal documents/laws into 2-3 pages and use the rest to research more cases (not 7,6 or 14 - you need much more and these need to be representative). Plus a theory and methodology is needed. Good luck!

Reviewer 2 Report

The topic of this manuscript is interesting; however, I recommend that the authors reorganize the manuscript to present their findings clearly. 

First, the materials and methods section needs to move up to right after the introduction (section 2). The section should describe clearly the search process for secondary materials (i.e. the news). What keywords were used? What search platforms were used? The procedure should be so clear that another researcher could take the same method and conduct the same study again. The authors mentioned selecting materials by three keywords, but this is different from the keywords they used when they first started the search. So please be sure to have both search keywords and selection criteria (inclusion keywords). Also, for inclusion keywords, please explain whether the authors looked for the keywords in the title, abstract, or the full text. 

The authors should clearly state the final sources they used for the analysis (i.e. for the results). For example, how many news articles did they have?

The authors described that they ended up with three cases from religious court decisions and three cases from district court decisions. Were these just the court documents? Or did they include relevant news from these 6 court cases? What documents pertaining to these 6 cases were analyzed? Did the authors only analyze these 6 cases and nothing else?  More descriptions are needed for these cases. 

In the method section, the authors should also explain their data analysis strategy, that is, how did they extract the content and come to the conclusions in the results section. Right now, the manuscript seems to be an editorial article where the authors cherry-pick what they want to discuss. The results would be more convincing if they could explain their analytic procedure and how they reached their conclusion.  

Reviewer 3 Report

The article topic is interesting. The writing flow is logical, but on the whole lacks the cohesiveness required for academic texts and has many limitations:

The article lacks a clear purpose and does not seem to be guided by a central idea. Readers may have difficulty understanding whether the text is theoretical or analytical in nature or systematic review. 

The manuscript should include a description of the methodology even if no empirical research has been conducted. In this case, in the context of discourse analysis.

The manuscript is not well contextualized in the existing literature in its current form. How is “Domestic Violence” being operationally defined in this paper? You should choose to include a more thorough literature review and expanded with current studies.

The problems with APA style and referencing in this manuscript were numerous. There was a considerable degree of mismatch between the in-text citations and the References section. Example the following references were not cited anywhere within the body of the paper: Bartky, Sandra;  Felstiner, William L.F., Richard L. Abel;  Galanter, Marc.

There are many errant commas scattered at inappropriate points in sentences throughout the text, as well as a misuse of semi-colons where commas or colons might be more appropriate.

Conclusion: This is a potentially interesting paper, but I think there are significant weaknesses in methodology and write-up.